# Caveolin-1 rs1997623 Single Nucleotide Polymorphism Creates a New Binding Site for the Early B-Cell Factor 1 That Instigates Adipose Tissue CAV1 Protein Overexpression

**DOI:** 10.3390/cells11233937

**Published:** 2022-12-06

**Authors:** Ashraf Al Madhoun, Dania Haddad, Rasheeba Nizam, Lavina Miranda, Shihab Kochumon, Reeby Thomas, Thangavel Alphonse Thanaraj, Rasheed Ahmad, Milad S. Bitar, Fahd Al-Mulla

**Affiliations:** 1Genetics and Bioinformatics, Dasman Diabetes Institute, Dasman 15462, Kuwait; 2Animal and Imaging Core Facilities, Dasman Diabetes Institute, Dasman 15462, Kuwait; 3Immunology & Microbiology Department, Dasman Diabetes Institute, Dasman 15462, Kuwait; 4Department of Pharmacology and Toxicology, Faculty of Medicine, Kuwait University, Jabriya 46300, Kuwait

**Keywords:** caveolin-1, rs1997623, metabolic syndrome, EBF1, AP-2α

## Abstract

Caveolin-1 (CAV1) is implicated in the pathophysiology of diabetes and obesity. Previously, we demonstrated an association between the *CAV1* rs1997623 C > A variant and metabolic syndrome (MetS). Here, we decipher the functional role of rs1997623 in *CAV1* gene regulation. A cohort of 38 patients participated in this study. The quantitative MetS scores (siMS) of the participants were computed. CAV1 transcript and protein expression were tested in subcutaneous adipose tissue using RT-PCR and immunohistochemistry. Chromatin immunoprecipitation assays were performed using primary preadipocytes isolated from individuals with different *CAV1* rs1997623 genotypes (AA, AC, and CC). The regulatory region flanking the variant was cloned into a luciferase reporter plasmid and expressed in human preadipocytes. Additional knockdown and overexpression assays were carried out. We show a significant correlation between siMS and *CAV1* transcript levels and protein levels in human adipose tissue collected from an Arab cohort. We found that the *CAV1* rs1997623 A allele generates a transcriptionally active locus and a new transcription factor binding site for early B-cell factor 1 (EBF1), which enhanced CAV1 expression. Our in vivo and in vitro combined study implicates, for the first time, EBF1 in regulating *CAV1* expression in individuals harboring the rs1997623 C > A variant.

## 1. Introduction

Caveolin-1 (CAV1) is an integral protein of caveolae. Caveolae are cholesterol-rich, smooth invaginations of the plasma membrane that partition into raft fractions. They are implicated in several essential cellular functions, such as endocytosis, transcytosis, maintenance of plasma membrane integrity, lipid homeostasis, signal transduction, and mechanoprotection [1]. The roles of caveolae and CAV1 in physiological and pathophysiological homeostasis have been extensively discussed [1,2]. CAV1 expression in adipose tissues (AT) is augmented in obese individuals with or without diabetes [3]. Moreover, elevated levels of CAV1 expression are seen in individuals with high abdominal circumference and obesity compared to healthy individuals [4].

The *CAV1* gene is 36 kb long and has three exons that produce eight transcripts. The default transcript, which is biologically relevant and highly conserved, encodes an active 178 aa protein. Multiple long noncoding RNAs are transcribed upstream or downstream of the *CAV1* gene, but their effects on CAV1 gene regulation or expression are unknown at this time. *CAV1* is differentially regulated depending on the cell type. Boopathi et al. found that GATA-6 might exert transcriptional repression of *CAV1* gene expression in human samples and murine models of bladder smooth muscle hypertrophy [5]. Park et al. showed that prolactin, via a Ras-dependent mechanism, negatively regulates *CAV1* gene expression in the mammary gland during lactation [6]. TLR signaling was also shown to be important for *CAV1* gene expression in human umbilical vein endothelial cells [7]. *CAV1* expression can also be regulated by a variety of microRNAs. MiR-124a has been shown to activate CAV1 in podocytes [8], miR-375 to downregulate *CAV1* gene expression, and miR-204 to target *CAV1* mRNA and decrease its expression in endothelial cells [9]. *CAV1* was identified as a target gene for miR103 and miR107 by Trajkovski et al. [10]. Their inactivation increases CAV1 expression in adipocytes, which improves IR maintenance, insulin signaling, adipocyte size, and glucose uptake.

Little is known about CAV1 regulation in adipocytes; single nucleotide polymorphisms (SNPs) in the *CAV1* gene have been associated with metabolic syndrome (MetS), which is a major risk factor for diabetes and coronary artery disease. Several *CAV1* SNPs were found to be associated with MetS, including rs926198 in Caucasians and Hispanics [11], rs3807989 in Han Chinese [12], rs11773845 in Latin Americans [13], rs3807992 in Persians [14], and rs1997623 in Kuwaiti children, as shown in our previous study [15], and as found in another adult population [16]. Although central obesity is a landmark trait for the diagnosis of MetS, the full spectrum of traits relating to MetS includes various other metabolic factors related to hypertension, diabetes, and dyslipidemia. Thus, evaluating the levels of CAV1 protein expression in individuals with varying MetS scores, and not just central obesity, could be insightful.

Herein, we suggest how a single nucleotide change in the 5′ untranslated region of *CAV1* can affect lipid metabolism. The variant rs1997623 is located within transcription factors’ binding sites in the upstream regulatory region of the *CAV1* gene. According to in silico analysis, this locus is targeted by multiple transcription factors, with the top two hits being activating protein-2 alpha (AP-2α) and early B-cell factor 1 (EBF1). Aside from the fact that AP-2α is expressed in preadipocytes and is suppressed during adipogenesis [17], little is known about its role in obesity or the metabolic syndrome. Meanwhile, EBF1 is known to play an important role in lipid metabolism [18] and is highly expressed in adipose tissue [19], but no link between EBF1 and CAV1 has ever been established prior to this study.

None of the earlier studies have attempted to elucidate the mechanism by which *CAV1* variants induce MetS; therefore, in this study, we first evaluated the expression levels of CAV1 in AT from adult individuals with differing MetS scores. Second, we investigated the mechanism by which the rs1997623 SNP regulates *CAV1* gene expression using functional assays for transcription factor binding, chromatin accessibility, and gene knockdown.

## 2. Materials and Methods

### 2.1. Ethics Statement

The study protocol was approved by the Ethical Review Committee of the Dasman Diabetes Institute, ensuring compliance with the guidelines of the Declaration of Helsinki and the US Federal Policy for the Protection of Human Subjects (project # RA CB-2021-007). Written informed consent was obtained from the participants prior to the collection of blood samples. The identities of the participants were protected from public exposure, and samples and data were processed anonymously.

### 2.2. Collection of Subcutaneous Adipose Tissue Biopsies

A cohort of 38 individuals participated in this study. AT samples (approximately 500 mg) were collected via abdominal subcutaneous fat pad biopsy using the standard surgical procedure, as described previously [20,21]. Briefly, the periumbilical area was decontaminated using alcohol gauze and was then locally anesthetized by injecting 2 mL of 2% lidocaine. A subcutaneous AT sample was collected through a small skin incision of 0.5 cm in length. The collected fat sample was further incised, rinsed in cold phosphate-buffered saline (PBS), fixed with 4% paraformaldehyde (Fisher Scientific, Pittsburgh, PA, USA) for 24 h, and embedded in paraffin or processed for DNA extraction using the DNAeasy Blood and Tissue Kit (Qiagen, Hilden, Germany). Simultaneously, freshly collected fat samples of 50–100 mg each were also preserved in RNAlater (Invitrogen, Burlington, ON, Canada) and stored at −80 °C until use [22].

### 2.3. Derivation of MetS Status and siMS Score

We adopted the adult diagnosis criteria defined by the International Diabetes Federation [23] to derive the MetS status of every study participant. To be diagnosed with MetS, the patients were required to have central obesity along with two or more of the following criteria: high triglycerides (TG), low high-density lipoprotein (HDL), hypertension, or a raised fasting blood glucose (FBG) level. An individual was considered to have (i) central obesity if the waist circumference (WC) was ≥94 cm in men or ≥80 cm in women; (ii) high TG if the serum TG level was ≥1.7 mmol/L (151 mg/dL), or if the patient was taking medication to lower TG; (iii) low HDL if the serum HDL cholesterol level was <1.03 mmol/L (40 mg/dL) in men or <1.29 mmol/L (50 mg/dL) in women, or if the patient was taking medication to restore HDL levels; (iv) hypertension if systolic blood pressure (SBP) was ≥130 mm Hg, diastolic blood pressure was ≥85 mm Hg, or if the patient was undergoing treatment for hypertension; and (v) diabetes if FBG was ≥5.6 mmol/L (100 mg/dL), or if the patient had been previously diagnosed with type 2 diabetes. Furthermore, we calculated the siMS score, which is a simple and accurate method for quantifying MetS in adults, using the formula described by Soldatovic [24]: siMS score = 2 × WC/Height + FBG/5.6 + TG/1.7 + SBP/130-HDL/1.02 for men, or HDL/1.28 for women.

### 2.4. Immunohistochemistry

Paraffin-embedded sections (4 μm thick) of subcutaneous AT were deparaffinized in xylene and rehydrated through descending grades of ethanol (100%, 95%, and 75%) to water. Antigen retrieval was performed by placing slides in a target retrieval solution (pH 6.0; Dako, Glostrup, Denmark) in a pressure cooker, boiling for 8 min, and cooling for 15 min. After washing in PBS, endogenous peroxidase activity was blocked with 3% H_2_O_2_ for 30 min, and nonspecific antibody binding was blocked with 5% fat-free milk for 1 h, followed by 1% bovine serum albumin solution for 1 h. The slides were incubated at room temperature overnight with primary antibody (1:100 dilution of rabbit polyclonal anti-Caveolin-1 antibody; Cell Signaling Technology #3238s). After washing with PBS (0.5% Tween), slides were incubated for 1 h with a secondary antibody, namely, goat anti-rabbit conjugated with horseradish peroxidase (HRP) polymer chain Dako EnVision Kit from (Dako, Glostrup, Denmark) and color was developed using 3,3′-diaminobenzidine (DAB) chromogen substrate. Specimens were washed, counterstained, dehydrated, cleared, and mounted, as described elsewhere [25]. For analysis, digital photomicrographs of the entire AT sections (20X; PannoramicScan, 3DHistech, Hungary) were used to quantify the immunohistochemical staining using ImageJ software (NIH, Bethesda, MD, USA).

### 2.5. RNA Extraction, cDNA Synthesis, and Reverse Transcription-Quantitative Polymerase Chain Reaction (RT-qPCR)

Total RNA was extracted from AT using the RNeasy kit (Qiagen, Valencia, CA, USA), as per the manufacturer’s instructions. cDNA was synthesized from 0.5 µg of RNA using the High-Capacity cDNA Reverse Transcription Kit (Applied Biosystems, Waltham, MA, USA). Real-time RT-qPCR was performed as previously described [26,27,28]. cDNA samples (50 ng) were amplified using TaqMan Gene Expression Master Mix (Applied Biosystems, Foster City, CA, USA) and gene-specific 20× TaqMan gene expression assays (Cav1, Hs00971716_m1; EBF1, Hs01092694_m1; GAPDH, Hs03929097_g1; Applied Biosystems, Foster City, CA, USA) on a 7500 Fast Real-Time PCR System (Applied Biosystems). Relative gene expression was calculated using the comparative cycle threshold (C_T_) method, as previously described [20,29]. Results were normalized to glyceraldehyde 3-phosphate dehydrogenase (GAPDH). Mean ± standard error of the mean (SEM) values are shown expressed as fold changes in expression relative to controls, as indicated [30].

### 2.6. DNA Constructs and Mutagenesis

Human DNA fragments of the *CAV1* regulatory fragment locus (250 bp) were amplified using PCR with the Advantage–GC cDNA PCR kit (Clontech, TakaraBio, Mountain View, CA, USA). They were sub-cloned into the appropriate restriction sites of the eukaryotic expression plasmids pRMT-Luc reporter vector (Origene, Rockville, MD, USA) and were verified by Sanger sequencing. The mutant relative to the *CAV1* rs1997623 variant was generated by site-directed mutagenesis (Stratagene, La Jolla, CA, USA) following protocols previously described [31,32].

### 2.7. Cell Culture, DNA Transfections, and Reporter Assays

Preadipocytes, isolated from individuals with AA, CC, or AC genotypes, were purchased from (ZenBio, New York, NC, USA). Cells were plated at a density of 2000 cells/well in 96-well plates and were transiently transfected using lipofectamine LTX (ThermoFisher, Waltham, MA, USA) with pRMT-Luc-CAV1-C, pRMT-Luc-CAV1-A, or pRMT-Luc control vector with or without an EBF1 mammalian expression vector (Origene, Rockville, MD, USA). Transfection efficiency was monitored by analyzing Renilla luciferase activity, as described previously [33]. After 24 h, the cells were washed twice with ice-cold PBS, lysed, and assayed as per the Dual-Luciferase Kit protocol (Promega, Madison, WI, USA). Luciferase activity was detected using the GloMax Navigator Microplate Luminometer (Promega, Madison, WI, USA) assay. The value denoting the activity of the luciferase reporter vector in the presence of EBF1 was normalized by subtracting the value denoting the activity of the luciferase reporter alone, in the absence of EBF1. Relative luciferase units (RLUs) represent luciferase activity normalized to Renilla activity [34].

### 2.8. Chromatin Immunoprecipitation (ChIP) Assays

ChIP assays were performed as described previously [32], with the following minor modifications: Preadipocytes were seeded on three 175 mm tissue culture flasks at 0.5 × 10^6^ cells per flask. After 48 h, 50 mg of chromatin from each cell line was immunoprecipitated using 0.5 ug of anti-AP-2α antibodies (C83E10, 3215, Cell Signaling, Danvers, MA, USA), anti-EBF1 specific antibodies (ab221033, Abcam, Branford, CT, USA), anti-Histone-3 acetyl K14 antibodies (H3K14^ac^, ab176799, Abcam, Branford, CT, USA), or nonspecific Rabbit IgG control antibodies (ab172730, Abcam, Branford, CT, USA). Briefly, cells were cross-linked with 4% formaldehyde and sonicated using a sonic dismembrator for a total of 30 times with 15-s pulses (1-min rest between the pulses), and lysates were cleared by centrifugation at 13,000 rpm for 30 min at 4 °C. The sheared chromatins were incubated with the described antibodies, and the immune complexes were captured using protein G-sepharose Dynabeads (Invitrogen, Oslo, Norway), as described previously [31]. AP-2α, EBF1, H3K14^ac^, or IgG-bound chromatins were quantified as a percent of chromatin input. Quantitative PCR (qPCR) analyses were performed using the forward 5′- GAGGTGAAGAGAAGCCAGGAAT-3′ and reverse 5′- CCCAATCTCAGGACCCCAAT-3′ primers. To be considered a true association, each ChIP sample was examined for the enrichment of a chromatin locus immunoprecipitated with a specific antibody and compared with the same chromatin locus immunoprecipitated with a nonspecific IgG (*p*-value < 0.05). Data are presented as mean ± SEM from three independent biological experiments.

### 2.9. Small Interfering RNA (siRNA) Transfection and Western Blotting

Before siRNA transfection, the pre-adipose cells were seeded on 6-well cell culture plates to achieve 70–80% confluency. After 24 h, combination EBF1 or scramble siRNAs (Santa Cruz, Dallas, TX, USA) were transfected with the lipofectamine LTX transfection reagent (1.5 μL/well). In each well, siRNAs were mixed to produce a final concentration of 100 pmol/well. Appropriate controls for treatment and nontreatment were also included, followed by 24 h of incubation. Total RNA was isolated and real-time qPCR was performed, as previously described, using the following primers: EBF1 Primers 5′-GTTTGTGGGGTTCGTGGAGA-3′, and 5′-CTGGGTTCTTGTCTTGGCCT-3′; CAV1 primers 5′-CGACCCTAAACACCTCAACGATG-3′ and 5′-GCAGACAGCAAGCGGTAAAACC-3′; and GAPDH1 primers 5′-CACTGGCGTCTTCACCACCATG-3′ and 5′-GCTTCACCACCTTCTTGATGTCA-3′.

Western blot analysis was performed, as described previously [30]. Briefly, cells were harvested and lysed in RIPA buffer. Cell lysates were quantified using the Quickstart Bradford assay (Bio-Rad Laboratories, Hercules, CA, USA), and equal amounts of protein were resolved on 8–12% polyacrylamide gels and transferred to nitrocellulose membranes (Bio-Rad Laboratories, Hercules, CA, USA). After blocking, the membranes were blotted with the following primary antibodies: EBF1 (ab108369, Abcam, Branford, CT, USA), CAV1 (3238s, Cell Signaling, Danvers, MA, USA), and b-actin (4970L, Cell Signaling, Danvers, MA, USA), and the corresponding HRP-linked secondary antibody (7074P2, Cell Signaling, Danvers, MA, USA). Proteins were visualized using the SuperSignal West Femto ECL kit (Thermo Scientific, Rockford, IL, USA). Images were captured using the ChemiDoc MP imaging system (Bio-Rad Laboratories, Hercules, CA, USA).

### 2.10. Pyrosequencing and Sanger Sequencing

Bisulfite conversion of genomic DNA (≥30 ng/µL) was performed using the EpiTect Fast Bisulfite Conversion Kit (Qiagen, Hilden, Germany), following the manufacturer’s protocol. Site-specific amplification of converted DNA was carried out using custom-designed *CAV1* primers (F1: AGGTGAACAGAAGTTAGGAATGTTT; R1: CCAATCTCAAAACCCCAATCC; sequencing primer: AAGTTAGGAATGTTTTATGT), and a Pyromark PCR kit on a thermal cycler programmed for an initial denaturation of 15 min at 95 °C, 45 cycles of (94 °C for 30 s, 56 °C for 30 s, and 72 °C for 30 s), followed by a final extension of 10 min at 72 °C. The resulting biotinylated PCR product (10 µL) and magnetic beads (3 µL) were pipetted into Q48 discs, and pyrosequencing was performed using the PyroMark Q48 Autoprep (Qiagen, Hilden, Germany), as per the manufacturer’s instructions. The same converted DNA was amplified and sequenced using conventional Sanger sequencing (ABI Prism 3730XL Genetic Analyzer, Applied Biosystems, Foster City, CA, USA), as previously described [35].

## 3. Results

### 3.1. MetS Score Correlates with CAV1 Transcript and Protein Levels in Human AT

We aimed to examine whether in vivo CAV1 expression is associated with MetS; therefore, we evaluated the mRNA and protein levels of *CAV1* in subcutaneous ATs isolated from 38 individuals with differing MetS scores (siMS). As observed in Figure 1A, a moderately positive and statistically significant correlation was observed between *CAV1* mRNA expression levels in AT and the MetS score status of the individuals studied. Using immunohistochemistry, we demonstrated that CAV1 proteins are in parallel with their mRNA expression in subcutaneous ATs isolated from individuals with a high siMS score [2,3,4] and a BMI > 30 (obese), as compared to individuals with a low siMS score [1,2] and a BMI < 25 (lean) (Figure 1B,C). Unfortunately, we did not have enough material from these individuals to perform SNP genotyping. Therefore, we directly proceeded to functionally assess the role of SNPs in *CAV1* expression.

### 3.2. CAV1 rs1997623 Alters the Transcription Factor Binding Site Motif and Mediates EBF1 Binding

Given the potential significance of these findings, we studied the association between the *CAV1* variant rs1997623 and *CAV1* transcriptional regulation in adipocytes in greater detail. It is worth noting that the transcriptional regulation of *CAV1* is complex. Locus mapping analysis revealed that the *CAV1* rs1997623 C > A variant spans the *CAV1* proximal regulatory region (Appendix A). Therefore, we hypothesized that the rs1997623 variant modifies *CAV1* gene regulation by altering the transcription factor binding site.

To identify the transcriptional regulation factor(s), we examined the immediate 21 bp flanking the variant locus using the bioinformatic database for transcription factor finder (TFBIND INPUT) [36]. The software predicted that the transcription factor AP-2α was a likely candidate that binds to the cis-element spanning the rs1997623-C wild-type variant (Appendix A). Next, we evaluated the binding of AP-2α to the wild-type rs1997623-C or the variant rs1997623-A using ChIP performed on preadipocytes isolated from individuals with either homozygous AA, homozygous wild-type CC, or heterozygous AC genotypes at the rs1997623 locus. Antibodies directed against AP-2α showed 8-fold chromatin enrichments in cells with the CC genotype relative to nonspecific IgG (Figure 2A). In contrast, no precipitation was observed using chromatins from cells with AA and AC genotypes, indicating the absence of AP-2α protein binding. We then wondered if any other transcription factor favored the binding of the cis-element spanning rs1997623-A. Similar bioinformatics analyses suggested EBF1 as a possible candidate (Appendix A). We tested this premise via ChIP analysis using a specific antibody directed against EBF1. The data showed significant enrichment in chromatin from cells with AA or AC genotypes but not the wild-type CC (Figure 2B). Collectively, these results suggest that EBF1 replaced AP-2α in the presence of the rs1997623-A variant.

The binding of either of the transcription factors to the corresponding DNA cis-element indicates a transcriptionally active site. Therefore, we next determined if histone acetylation levels were changed at the rs1997623 locus. ChIP assays were performed with antibodies against acetylated lysine 14 at histone 3 (H3K14^ac^), indicative of actively transcribed chromatin. Notably, the binding of AP-2α to the rs1997623-C locus or EBF1 to the rs1997623-A locus correlated with elevated H3K14 acetylation, suggesting a putative active transcription in this region (Figure 2C). We then tested CAV1 promoter methylation by pyrosequencing; the analysis showed nonsignificant (2–5%) methylation independent of the genotype. This indicates that the rs1997623-C locus is unmethylated and excludes this cytosine as a methylation-based regulated nucleotide (Appendix A).

### 3.3. CAV1 rs1997623 Causes Induced Promoter Activity at its Locus

Next, we studied the effect of this SNP on promotor activation by cloning 250 bp of the *CAV1* regulatory region flanking the rs1997623 variant into a luciferase reporter construct (named wild-type, CAV1-C-Luc) and generating a single mutation C > A (named CAV1-A-Luc) equivalent to the CAV1 rs1997623-A variant, as described in the Materials and Methods section (Figure 3A). Preadipocytes were transfected with CAV1-C-Luc or CAV1-A-Luc constructs separately. As shown in Figure 3B, luciferase reporter activity was detected in both constructs (dashed bars); however, the expression was significantly higher in CAV1-A-Luc (1.6-fold relative to CAV1-C-Luc, *p* = 0.003). To demonstrate the influence of EBF1 on driving the expression further, we co-transfected CAV1-C-Luc or CAV1-A-Luc with constructs that overexpressed EBF1. Our data showed a 2.5-fold increase in the reporter activity of construct CAV1-A-Luc compared to CAV1-C-Luc alone (*p* = 0.001; Figure 3B, black bars).

Taken together, the above results indicate that the promoter located in the 250 bp of the *CAV1* regulatory region is transcriptionally active in preadipocytes and that its activity is further enhanced in the presence of the *CAV1* rs1997623-A construct that attracted endogenous or overexpressed exogenous EBF1.

### 3.4. EBF1 siRNA Reduces the Expression of CAV1 in PreAdipocytes

Having shown that EBF1 binds the rs1997632-A allele and induces a more rigorous expression, we next sought to determine whether silencing EBF1 could reduce *CAV1* gene expression in the preadipocyte model. To investigate this, we studied the expression of CAV1 before and after lowering EBF1 levels using siRNA technology (Figure 4). Transfection with EBF1 siRNA reduced both EBF1 and subsequently the *CAV1* gene transcripts by 50–60% compared to the control cells (Figure 4A). These data were further confirmed using Western blot analysis. EBF1 siRNA specifically reduced CAV1 expression in preadipocytes with AA and AC genotypes at the *CAV1* rs1997623 variant locus, as shown in Figure 4B,C, implying a direct association between EBF1 and CAV1 expression levels in these cells.

## 4. Discussion

CAV1 is known to regulate hepatic lipid accumulation, lipid and glucose metabolism, mitochondrial energy homeostasis, and hepatocyte proliferation by modulating several molecular pathways [37]. Thus, the dysregulated expression of CAV1 has several avenues that could lead to MetS. Our study examines, for the first time, the promoter regulatory feature of the rs1997623 variant of *CAV1* through functional assays. Our results demonstrate that the *CAV1* SNP, located in the gene regulatory region, alters the transcription consensus site to favor EBF1 binding. Moreover, our functional study indicated an increase in promoter activity through the generation of an active binding site for the transcription factor EBF1 and sustained active transcription by maintaining open chromatin for the recruitment of EBF1 to the *CAV1* upstream regulatory region in human adipocytes. Our findings are summarized in Figure 5.

As per the Ensembl genome annotation system [38], the studied variant overlaps nine transcript isoforms of *CAV1* (Appendix A). CAV1 has five protein isoforms of different lengths, ranging from 86 to 178 amino acids (Appendix A). Depending on the transcript and protein isoform, rs1997623 is treated as a missense (leading to amino acid change), synonymous (leading to no change in amino acid) mutation, or an intronic or an upstream gene variation. Further, the Ensembl genome annotation system annotates the studied variant as a regulatory region variant within the promoter (Appendix A).

Previously, EBF1 has been implicated in promoting adipogenesis and controlling genes involved in terminal adipocyte differentiation [39]. These functional insights into the involvement of *CAV1* genetic variant rs1997623 further strengthen the concept that CAV1 contributes to MetS susceptibility and represents an attractive molecular target for the prevention and treatment of MetS. The mechanisms behind the contribution of CAV1 protein to MetS are poorly understood. An attractive model that we have previously shown suggests a link between CAV1 protein overexpression and the promotion of cellular senescence [40]. This model, along with others, is currently under investigation in our laboratory.

The following caveats should be considered when interpreting our results: First, we focused on the Kuwaiti population alone, and the results may not apply to other ethnic groups. Second, as shown in Figure 2, homozygous cells (rs1997623 AA) appear to bind less EBF1 than heterozygotes, which needs to be explained further. However, this phenotypic heterogeneity may stem from using preadipocytes from different individuals.

## 5. Conclusions

The current study was based on our previous discovery of a significant association between rs1997623-A and MetS in children [15]. We found that increasing MetS scores correlated with higher in vivo CAV1 expression levels in human AT. The regulation of the *CAV1* rs1997623-A locus in preadipocytes could be influenced, at least partly, by the recruitment and allele-specific binding of the EBF1 transcription factor.

## Figures and Tables

**Figure 1 cells-11-03937-f001:**
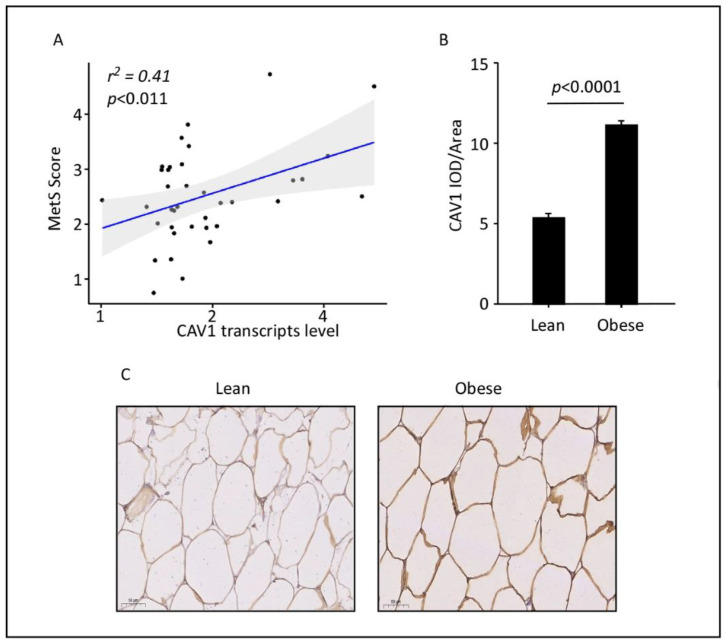
CAV1 transcripts and protein levels in adipose tissues. (**A**) Significant correlation between MetS score and *CAV1* transcript levels in the AT of 38 Kuwaiti individuals with varying siMS scores (r^2^ = 0.414, *p*-value < 0.011). (**B**) Integrated optical density (IOD) divided by the adipocyte area was used to quantify CAV1 protein expression in immunohistochemistry (IHC) sections of the adipose tissue samples from lean and obese individuals. The data (mean ± SEM) show significantly elevated levels of CAV1 proteins in subcutaneous ATs from obese versus lean individuals (*p* < 0.0001). Five individuals were considered for each of the obese (BMI: >30 Kg/m^2^) and lean (BMI: <25 Kg/m^2^) categories. (**C**) Representative immunohistochemistry images (100× magnification; scale bar 50 μm). The IHC analysis was performed to determine CAV1 protein expression using five AT sections isolated from five participants per group, each with similar results showing the comparative adipose CAV1 protein expression in lean and obese individuals.

**Figure 2 cells-11-03937-f002:**
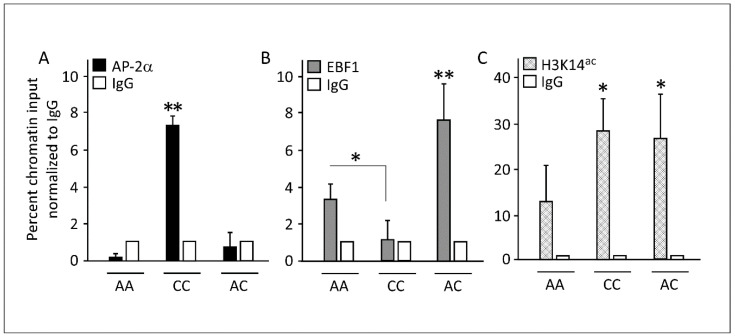
Binding of AP-2α and EBF1 to CC and AA genotypes, respectively, mediated allele-specific active transcription. (**A**) ChIP analysis of AP-2α at the CAV1 rs1997623 locus in preadipocytes isolated from three human individuals revealed that AP-2α specific antibodies are bound and enriched chromatins in cells with genotype CC but not AA or AC. The binding at CC was significantly higher compared to AA; however, the binding at AC was not significantly higher compared to AA. (**B**) ChIP analysis of EBF1 binding at the CAV1 rs1997623 locus revealed significant binding enrichments at both AA and AC genotypes. The binding at AC was significantly higher compared to AA, however, the binding at CC was significantly lower compared to AA. (**C**) ChIP analysis of H3K14ac-specific antibodies at the CAV1 rs1997623 locus indicated active transcription in all cells independent of their genotype, albeit at different rates. Significant enrichments were found in the CC and AC genotypes when compared to the AA genotype. White bars designate genomic regions precipitated with IgG-nonspecific antibodies. Percentage chromatin input was calculated using real-time qPCR analysis and the locus-specific primers. Relative enrichment was calculated as the percent chromatin input normalized to IgG from three biological replicas (*n* = 3). Two-tailed unpaired Student’s *t*-test was used to determine statistical significance. * *p* < 0.01, ** *p* < 0.001.

**Figure 3 cells-11-03937-f003:**
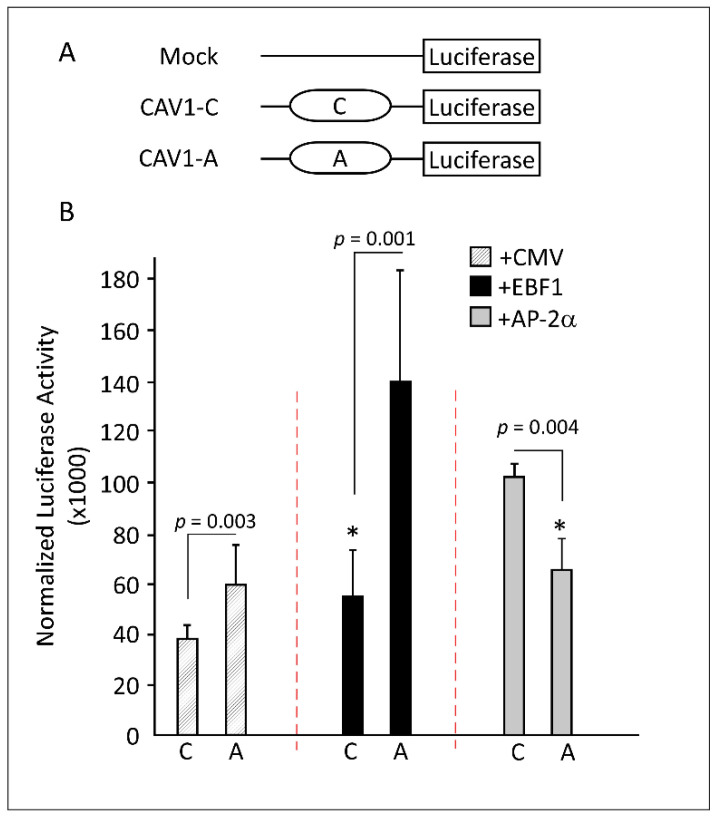
Promoter activity analysis using wild-type versus the rs1997623 variant luciferase reporter assays. (**A**) A schematic representation of the *CAV1* gene mapping the rs1997623 variant to intron 1; the cloned 250 bp DNA fragment spanning the *CAV1* indel rs1997623 into Luciferase-pRMT-Luc reporter vector as well as the generated C > A mutant are shown. (**B**) Luciferase reporter assay of rs1997623. Pre-adipose cells transfected with CAV1-C-Luc or CAV1-A-Luc constructs and co-transfected with control plasmid (shaded bars), EBF1 (black bars), or AP-2α (gray bars) constructs. Luciferase activities are presented as folds change relative to the pRMT-Luc reporter. Transfection efficiency was monitored by assaying the activity of co-transfected Renilla luciferase. *p*-values are given when comparing CAV1-C-Luc to CAV1-A-Luc transfections with or without EBF1 and AP-2α. * Indicates nonsignificance relative to the reciprocal CMV plasmid transfection. Data are presented as mean ± standard error of the mean (SEM) and the *p*-value of a two-tailed unpaired *t*-test.

**Figure 4 cells-11-03937-f004:**
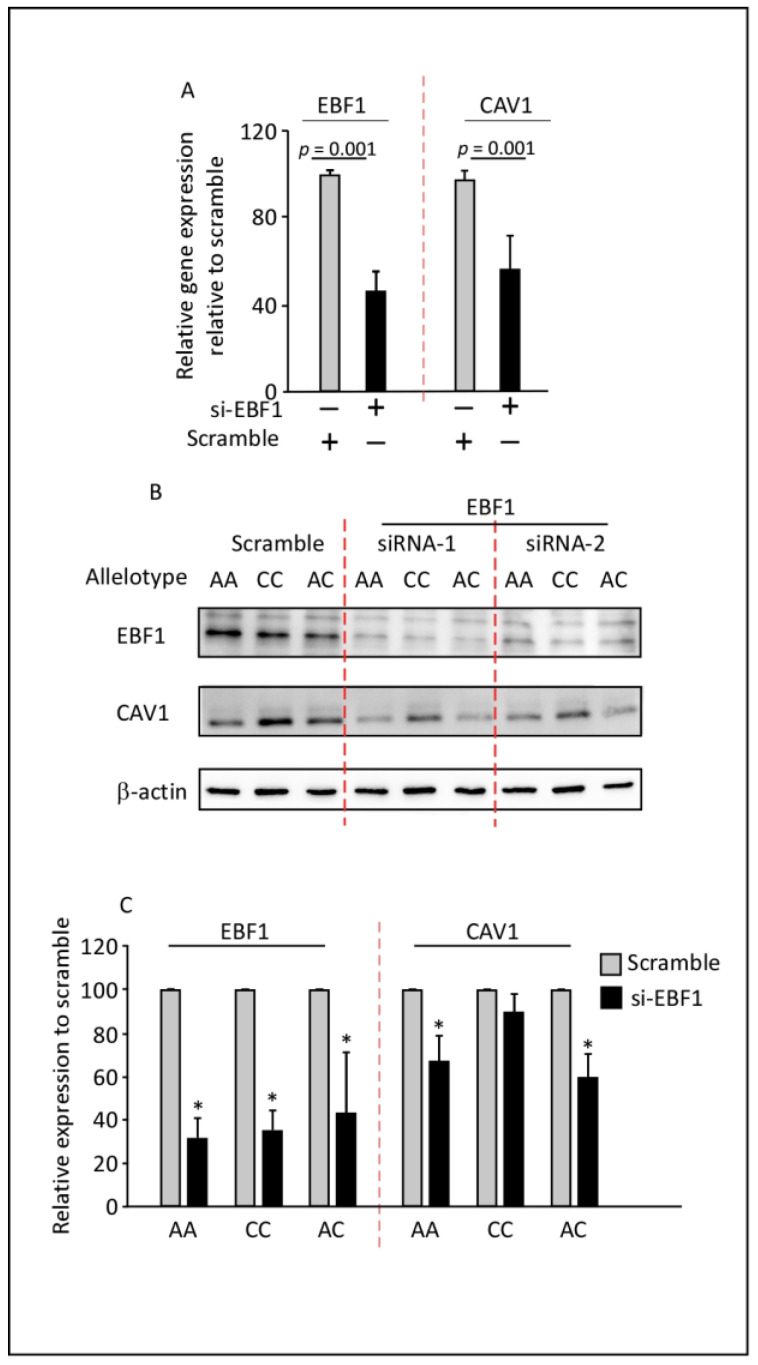
EBF1 siRNA reduces CAV1 expression levels in preadipocytes. (**A**) EBF1 knockdown using siRNA resulted in a significant reduction in CAV1 expression levels. RT-qPCR results were normalized to GAPDH and expressed relative to scramble siRNA controls. Data are presented as mean ± SD; unpaired two-tailed t-test (*n* = 3). (**B**,**C**) Preadipocytes were transiently transfected with scramble siRNA or with EBF1 siRNA. The Western blots were developed with antibodies against EBF1, CAV1, and β-actin, as indicated. EBF1 expression was significantly reduced in all genotypes. CAV1 expression was reduced in the preadipocytes harboring AA and AC genotypes compared to the scrambled control for each genotype. Data are presented as mean ± SD; unpaired two-tailed t-test (*n* = 3). CAV1: caveolin-1; EBF1: early B-cell factor 1; AA: *CAV1* rs1997623 variant A alleles; CC: *CAV1* rs1997623 variant C alleles; AC: *CAV1* rs1997623 variant with A and C alleles. * *p* < 0.01.

**Figure 5 cells-11-03937-f005:**
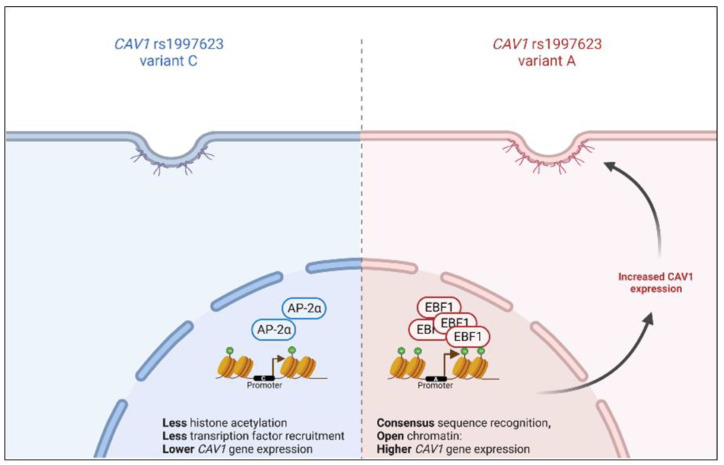
Mechanism of gene expression regulation by the *CAV1* rs1997623 variant. The CAV1 rs1997623 allele-specific transcription factor binding increases histone acetylation and introduces differential transcription factor binding, leading to enhanced CAV1 gene expression. AP-2α: activating protein-2 alpha; EBF1: early B-cell factor 1. Created with BioRender.com (accessed on 1 September 2020).

## Data Availability

Not applicable.

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
