# Peer review of "Caveolin-1 rs1997623 Single Nucleotide Polymorphism Creates a New Binding Site for the Early B-Cell Factor 1 That Instigates Adipose Tissue CAV1 Protein Overexpression"

_cells, 2022, doi:10.3390/cells11233937_

Round 1

Reviewer 1 Report

The authors of this manuscript are to be congratulated; this is a well written (minor typos) and interesting bit of research that may have potential clinical significance.

The novel observation relates to a mechanism by which a SNP in caveolin-1 creates a binding site for the transcription factor Early B Cell Factor 1 that results in caveolin-1 overexpression in adipocytes. The functional significance of this is not entirely clear, but correlates to a degree with quantitative metabolic syndrome scores (miMS). The molecular methods used to make the novel observation and validate the findings are robust, well detailed, and convincing.

Limitations that require consideration or correction include:

1.  Statistical analysis where Student's t-test was used for groups of 3 or more when ANOVA plus post hoc analysis should be employed

2. Literature review of the other transcription factors that are known to regulate CAV1 gene expression (e.g., NFkB, Egr1, etc) and whether EBCF1 activity on the CAV1 promoter induces a unique effect.

3. Is the rs19976243 SNP in the caveolin-1 promoter unique to adipocytes ? What does this do to Cav-1 expression in endothelial cells, fibroblasts, or smooth muscle cells?

4. Discussion of the functional significance of caveolin-1 overexpression in adipocytes which already express a lot of the protein could/should be expanded. Does this lead to greater TG or FFA uptake? Modulation of signaling in these cells. There is some evidence that overexpression of Cav1, which is known to negatively regulate signaling proteins like eNOS and other enzymes, ion channels, and receptors, may inhibit caveolae-mediated endocytosis. Is this case in adipocytes with the SNP? 

Author Response

The authors of this manuscript are to be congratulated; this is a well written (minor typos) and interesting bit of research that may have potential clinical significance.

The novel observation relates to a mechanism by which a SNP in caveolin-1 creates a binding site for the transcription factor Early B Cell Factor 1 that results in caveolin-1 overexpression in adipocytes. The functional significance of this is not entirely clear, but correlates to a degree with quantitative metabolic syndrome scores (miMS). The molecular methods used to make the novel observation and validate the findings are robust, well detailed, and convincing.

Limitations that require consideration or correction include:

  1. Statistical analysis where Student's t-test was used for groups of 3 or more when ANOVA plus post hoc analysis should be employed

Thank you for this comment, in fact we didn’t use ANOVA because we didn’t use more than 2 groups at a time. In Figure 2, C/C and A/C genotypes were compared to A/A genotype and only significant values were shown. In Figure 4-C, each genotype (AA, CC, and AC) was compared to its scrambled control and only significant values were shown. Unambiguous legends for these figures are added in the manuscript (Please see the legends for Figure 2 and 4 (page 7, lines 246-258; and page 10, lines 300-308.

Response:

  1. Literature review of the other transcription factors that are known to regulate CAV1 gene expression (e.g., NFkB, Egr1, etc) and whether EBCF1 activity on the CAV1 promoter induces a unique effect.

Thank you for the requested literature review, we have edited the manuscript, accordingly. Please see page 2, lines 40-52.

Please note that CAV1 regulates the expression of multiple proteins and responds to a variety of signaling pathways and stimuli. However, little is known about its regulation in adipocytes and no one has ever attempted to check what a SNP does to CAV1 expression and this is where our focus resides. NFKB binding scored lower than EBF1 or AP-2a according to our in-silico analysis, that is why we didn’t include it in our study design. There is no indication- thus far in literature that Egr1 regulates CAV1 expression besides in a non-peer reviewed abstract published in a conference in 2006 [AACR, 97th annual meeting, 2006, 66, 8_supplement, 31.]. On the contrary, all other published data indicate that CAV1 is needed for Egr1 expression and translocation. There is no previously published data linking EBF1 and CAV1 in any cellular type, we are the first ones finding that EBF1 can regulate CAV1 expression.

  1. Is the rs19976243 SNP in the caveolin-1 promoter unique to adipocytes ? What does this do to Cav-1 expression in endothelial cells, fibroblasts, or smooth muscle cells?

We thank the reviewer for the requested clarification. We did not study the effect of the CAV1 variant in cell types other than adipocytes. We focused only on adipocytes because they are the major cell type in the adipose tissue and are implicated in obesity and the metabolic syndrome which is the main disorder studied in this manuscript.

  1. Discussion of the functional significance of caveolin-1 overexpression in adipocytes which already express a lot of the protein could/should be expanded. Does this lead to greater TG or FFA uptake? Modulation of signaling in these cells. There is some evidence that overexpression of Cav1, which is known to negatively regulate signaling proteins like eNOS and other enzymes, ion channels, and receptors, may inhibit caveolae-mediated endocytosis. Is this case in adipocytes with the SNP? 

This SNP is located in the upstream regulatory region of CAV1 gene, not in the exonic region thus it does not alter CAV1 function; hence, we studied it solely in the context of CAV1 gene regulation and CAV1 expression. It will be interesting to investigate the role of CAV1 up-regulation relative to other signaling pathways, which may indeed cause the observed metabolic dysregulation in obesity. We have previously thoroughly discussed the role of CAV1 in the regulation of different receptors (GLUT4, IRS1), proteins (MDM2, sirt1, eNOS), and lipid homeostasis in our published review (Haddad et. al, 2020).

Reviewer 2 Report

This paper studies the role of rs1997623 in CAV1 gene expression.  The authors describe that metabolic syndrome (MetS) score and CAV1 expression in adipose tissues shows positive correlation. By analyzing the isolated pre-adipocytes cells from three different CAV1 rs1997623 genotype (AA, AC and CC), and found CAV1 rs1997623 A allele produces a EBF1 (Early B Cell Factor 1) binding site that increases CAV1 expression.  The manuscript contains interesting findings and has merit publication.  However, there are several points to be addressed before publication.

Major Comments:

1.     There is no description for EBF-1 and AP-2 alpha. The authors should describe those in introduction. 

2.     The authors mentioned "significant positive correlation" in line 212, but r^2 = 0.41 indicates that the linear line does not fit well with data in spite of the positive correlation at p<0.011. If the largest and smallest values are eliminated, is the conclusion same?

3.     Authors suddenly started to mention the variant, CAV1 rs1997623 in line 230. The rationale for the investigation about the variant is hard to understand.

4.     Fig. 2B: Why EBF1 binding of A/C was higher than A/A?  The authors should discuss possible reasons.

5.     Fig. 3:  EBF1 expression level seems to be different among the three samples.  CAV1 expression level is also different among the three samples.  The expression level of these proteins among the A/A, A/C, and C/C should be examined with several independent experiments for significance.  The expression level influences the data.

6.     The authors should show the effect of AP2 gene silencing.

7.     Fig. 3:  The authors should show the effect of AP2 on luciferase assay.

8.     It is interesting AP2 has no effect on A/C CHIP data, while AP2 showed very large effect on C/C data.  Please discuss the possible reasons. 

Minor Comments:

1.     Fig. 1.   Are the error bars SD or SE?

2.     Fig. 4. “A” in legend is missing.  n number of “C” is missing. SD or SE?

Author Response

This paper studies the role of rs1997623 in CAV1 gene expression.  The authors describe that metabolic syndrome (MetS) score and CAV1 expression in adipose tissues shows positive correlation. By analyzing the isolated pre-adipocytes cells from three different CAV1 rs1997623 genotype (AA, AC and CC), and found CAV1 rs1997623 A allele produces a EBF1 (Early B Cell Factor 1) binding site that increases CAV1 expression.  The manuscript contains interesting findings and has merit publication.  However, there are several points to be addressed before publication.

Major Comments:

  1. There is no description for EBF-1 and AP-2 alpha. The authors should describe those in introduction. 

Thank you for the comment, the introduction has been modified. Please see page 2 , lines 62-69.

  1. The authors mentioned "significant positive correlation" in line 212, but r^2 = 0.41 indicates that the linear line does not fit well with data in spite of the positive correlation at p<0.011. If the largest and smallest values are eliminated, is the conclusion same?

We thank the reviewer for this comment. It is true that our results only explain 41% (r^2 = 0.41) of the model, however the p-value is significant. This is possibly due to our limited adipose tissue samples (only 38) and the variability in CAV1 expression among individuals, In the future, we plan to collect and analyze a larger cohort number, however, the trend of the correlation between CAV1 expression and MetS would probably stay the same.

We removed the lowest 4 values, which are most likely presented with low MetS, as a result, the correlation was notable at r2 = 0.36 (p = 0.041). We should not remove the high values as they are individuals with true strong MetS.

  1. Authors suddenly started to mention the variant, CAV1 rs1997623 in line 230. The rationale for the investigation about the variant is hard to understand.

We have now modified the introduction in a way to better introduce the studied SNP. Please see page 2 , lines 62-74

  1. 2B: Why EBF1 binding of A/C was higher than A/A?  The authors should discuss possible reasons.

Response:

We are equally perplexed about why EBF1 binding of A/C was higher than A/A. We have repeated this experiment multiple times and it gave us the same trend. The only plausible explanation is that the cells with the A/C genotype might carry epigenetically modified chromatin or other proteins that facilitate EBF1 binding to this hybrid locus unlike the C/C cells. Furthermore, these cells are coming from two different individuals. Similarly, the cells with the C/C genotype showed 8-folds binding capacity of AP-2a relative to the A/C genotype. This indicates a possible implication of several epigenetic mechanisms that regulate CAV1.

  1. 3:  EBF1 expression level seems to be different among the three samples.  CAV1 expression level is also different among the three samples.  The expression level of these proteins among the A/A, A/C, and C/C should be examined with several independent experiments for significance.  The expression level influences the data.

Figure 3 is an in-vitro luciferase reported assay performed using only one type of pre-adipocyte, in which the endogenous levels of CAV1 and EBF1 are expected to be at the same levels. The reporter assay data were detected after transfections with a construct containing 250 bp of CAV1 regulatory region with either the re1997623-C or re1997623-A (generated by single nucleotide direct mutagenesis and confirmed with Sanger sequencing), with or without co-transfection with EBF1 or AP2 constructs.

  1. The authors should show the effect of AP2 gene silencing.

Thank you for the suggestion. We have also focused on EBF1 in this paper for the following reasons:

AP-2a  is a ubiquitous transcription factor that is involved in several pathways responsible for cell homeostasis. AP2 siRNA may result in global alternation that influences CAV1 indirectly and independently from its binding to the locus. However, EBF1 is an adipogenic transcription factor.

We overexpressed AP-2a  and studied CAV1 and other signaling pathways that have direct influences on CAV1 gene expression. The data indicate that CAV1 is differentially regulated in the presence or absence of AP-2a and in response to different microenvironmental conditions. The data are unrelated to the variant under consideration and are beyond the scope of this manuscript where we are showing the effect of one CAV1 SNP on CAV1 gene expression. Unshown AP-2a results will be polished; several proteins will be studied, and the results will be submitted in a different paper discussing CAV1 regulation in general without the role of an intrinsic polymorphism.

Fig. 3:  The authors should show the effect of AP2 on luciferase assay.

We thank the reviewer for their suggestion. AP-2a. Figure 3 has been modified to include the effect of AP-2a on luciferase assay. Please see page 9.

  1. It is interesting AP2 has no effect on A/C CHIP data, while AP2 showed very large effect on C/C data.  Please discuss the possible reasons.

It is true that one would expect to see the binding of AP-2a on AC to be half of CC but what we see in figure 2 is the result of multiple repeats. The used primary cell lines are originally from different individuals, so there could be epigenetic modifications that are affecting AP-2a binding in the A/C cell line.

 Minor Comments:

  1. 1.   Are the error bars SD or SE?

We thank the reviewer for bringing this to our notice. The figure legend has been corrected. Please see page 6, lines 223.

  1. 4. “A” in legend is missing.  n number of “C” is missing. SD or SE?

We thank the reviewer for bringing this to our notice. The figure legend has been corrected. Please see page 10, lines 322-323.
